# Rethinking Space-Time Networks with Improved Memory Coverage for Efficient Video Object Segmentation

**Ho Kei Cheng**[†]
University of Illinois
Urbana-Champaign
hokeikc2@illinois.edu

**Yu-Wing Tai**
Kuaishou
Technology
yuwing@gmail.com

**Chi-Keung Tang**
The Hong Kong University of
Science and Technology
cktang@cs.ust.hk

## Abstract

This paper presents a simple yet effective approach to modeling space-time correspondences in the context of video object segmentation. Unlike most existing approaches, we establish correspondences directly between frames without re-encoding the mask features for every object, leading to a highly efficient and robust framework. With the correspondences, every node in the current query frame is inferred by aggregating features from the past in an associative fashion. We cast the aggregation process as a voting problem and find that the existing inner-product affinity leads to poor use of memory with a small (fixed) subset of memory nodes dominating the votes, regardless of the query. In light of this phenomenon, we propose using the negative squared Euclidean distance instead to compute the affinities. We validate that every memory node now has a chance to contribute, and experimentally show that such diversified voting is beneficial to both memory efficiency and inference accuracy. The synergy of correspondence networks and diversified voting works exceedingly well, achieves new state-of-the-art results on both DAVIS and YouTubeVOS datasets while running significantly faster at 20+ FPS for multiple objects without bells and whistles.

## 1  Introduction

Video object segmentation (VOS) aims to identify and segment target instances in a video sequence. This work focuses on the semi-supervised setting where the first-frame segmentation is given and the algorithm needs to infer the segmentation for the remaining frames. This task is an extension of video object tracking [1, 2], requiring detailed object masks instead of simple bounding boxes. A high-performing algorithm should be able to delineate an object from the background or other distractors (e.g., similar instances) under partial or complete occlusion, appearance changes, and object deformation [3].

Most current methods either fit a model using the initial segmentation [4, 5, 6, 7, 8, 9] or leverage temporal propagation [10, 11, 12, 13, 14, 15, 16], particularly with spatio-temporal matching [17, 18, 19, 20, 21, 22, 23, 24, 25, 26, 27]. Space-Time Memory networks [18] are especially popular recently due to its high performance and simplicity – many variants [22, 16, 23, 21, 24, 28, 29, 30], including competitions' winners [31, 32], have been developed to improve the speed, reduce memory usage, or to regularize the memory readout process of STM.

In this work, we aim to subtract from STM to arrive at a minimalistic form of matching networks, dubbed Space-Time Correspondence Network (STCN) [1]. Specifically, we start from the basic premise

---

[†]This work was done in The Hong Kong University of Science and Technology.
[1]Training/inference code and pretrained models: https://github.com/hkchengrex/STCN

35th Conference on Neural Information Processing Systems (NeurIPS 2021).

that *correspondences are target-agnostic*. Instead of building a specific memory bank and therefore affinity for every object in the video as in STM, we build a single affinity matrix using only RGB relations. For querying, each target object passes through the same affinity matrix for feature transfer. This is not only more efficient but also more robust – the model is forced to learn all object relations beyond just the labeled ones. With the learned affinity, the algorithm can propagate features from the first frame to the rest of the video sequence, with intermediate features stored as memory.

While STCN already reaches state-of-the-art performance and speed in this simple form, we further probe into the inner workings of the construction of affinities. Traditionally, affinities are constructed from dot products followed by a softmax as in attention mechanisms [18, 33]. This however implicitly encoded "confidence" (magnitude) with high-confidence points dominating the affinities all the time, regardless of query features. Some memory nodes will therefore be always suppressed, and the (large) memory bank will be underutilized, reducing effective diversity and robustness. We find this to be harmful, and propose using the negative squared Euclidean distance as a similarity measure with an efficient implementation instead. Though simple, this small change ensures that every memory node has a chance to contribute significantly (given the right query), leading to better performance, higher robustness, and more efficient use of memory.

Our contribution is three-fold:

- We propose STCN with direct image-to-image correspondence that is simpler, more efficient, and more effective than STM.

- We examine the affinity in detail, and propose using L2 similarity in place of dot product for a better memory coverage, where every memory node contributes instead of just a few.

- The synergy of the above two results in a simple and strong method, which suppresses previous state-of-the-art performance without additional complications while running fast at 20+ FPS.

## 2   Related Works

**Correspondence Learning**   Finding correspondences is one of the most fundamental problems in computer vision. Local correspondences have been used heavily in optical flow [34, 35, 36] and object tracking [37, 38, 39] with fast running time and high performance. More explicit correspondence learning has also been achieved with deep learning [40, 41, 42].

Few-shots learning can be considered as a matching problem where the query is compared with every element in the support set [43, 44, 45, 46]. Typical approaches use a Siamese network [47] and compare the embedded query/support features using a similarity measure such as cosine similarity [43], squared Euclidean distance [48], or even a learned function [49]. Our task can also be formulated as a few-shots problem, where our memory bank acts as the support set. This connection helps us with the choice of similarity function, albeit we are dealing with a million times more pointwise comparisons.

**Video Object Segmentation**   Early VOS methods [4, 5, 50] employ online first-frame finetuning which is very slow in inference and have been gradually phased out. Faster approaches have been proposed such as a more efficient online learning algorithm [8, 6, 7], MRF graph inference [51], temporal CNN [52], capsule routing [53], tracking [11, 13, 15, 54, 55, 56, 57], embedding learning [10, 58, 59] and space-time matching [17, 18, 19, 20]. Embedding learning bears a high similarity to space-time matching, both attempting to learn a deep feature representation of an object that remains consistent across a video. Usually embedding learning methods are more constrained [10, 58], adopting local search window and hard one-to-one matching.

We are particularly interested in the class of Space-Time Memory networks (STM) [18] which are the backbone for many follow-up state-of-the-art VOS methods. STM constructs a memory bank for each object in the video, and matches every query frame to the memory bank to perform "memory readout". Newly inferred frames can be added to the memory, and then the algorithm propagates forward in time. Derivatives either apply STM at other tasks [21, 60], improve the training data or augmentation policy [21, 22], augment the memory readout process [16, 21, 22, 24, 28], use optical flow [29], or reduce the size of the memory bank by limiting its growth [23, 30]. MAST [61] is an adjacent research that focused on unsupervised learning with a photometric reconstruction loss. Without the input mask, they use Siamese networks on RGB images to build the correspondence out of necessity. In this work, we deliberately build such connections and establish that building

correspondences between images is a better choice, even when input masks are available, rather than a concession.

We propose to overhaul STM into STCN where the construction of affinity is redefined to be between frames only. We also take a close look at the similarity function, which has always been the dot product in all STM variants, make changes and comparisons according to our findings. The resultant framework is both faster and better while still principled. STCN is even fundamentally simpler than STM, and we hope that STCN can be adopted as the new and efficient backbone for future works.

## 3  Space-Time Correspondence Networks (STCN)

Given a video sequence and the first-frame annotation, we process the frames sequentially and maintain a memory bank of features. For each query frame, we extract a **key** feature which is compared with the keys in the memory bank, and retrieve corresponding **value** features from memory using key affinities as in STM [18].

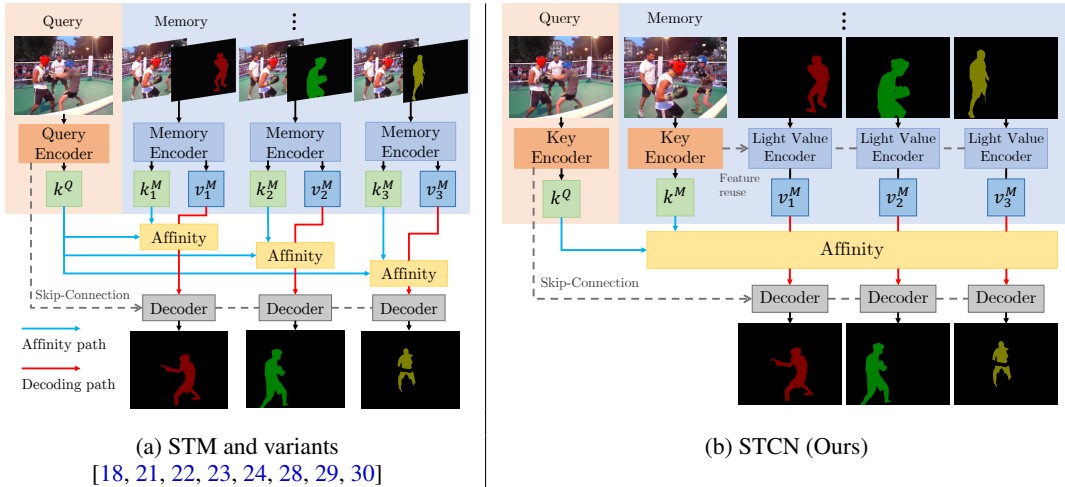

(a) STM and variants
[18, 21, 22, 23, 24, 28, 29, 30]

(b) STCN (Ours)

Figure 1: **Left**: The general framework of the popularly used Space-Time Memory (STM) networks, ignoring fine-level variations. Objects are encoded separately, and affinities are specific to each object. **Right**: Our proposed Space-Time Correspondence Networks (STCN). We use Siamese key encoders to compute affinity directly from RGB images, making it more robust and efficient. Note that the query key can be cached and reused later as a memory key (unlike in STM) as it is independent of the mask.

### 3.1  Feature Extraction

Figure 1 illustrates the overall flow of STCN. While STM [18] parameterizes a *Query Encoder* (image as input) and a *Memory Encoder* (image and mask as input) with two ResNet50 [62], we instead construct a *Key Encoder* (image as input) and a *Value Encoder* (image and mask as input) with a ResNet50 and a ResNet18 respectively. Thus, unlike in STM [18], the **key** features (and thus the resultant affinity) can be extracted independently without the mask, computed only once for each frame, and symmetric between memory and query.[2] The rationales are 1) Correspondences (key features) are more difficult to extract than value, hence a deeper network, and 2) Correspondences should exist between frames in a video, and there is little reason to introduce the mask as a distraction. From another perspective, we are using a Siamese structure [47] which is widely adopted in few-shots learning [63, 49] for computing the key features, as if our memory bank is the few-shots support set. As the key features are independent of the mask, we can reuse the "query key" later as a "memory key" if we decide to turn the query frame into a memory frame during propagation (strategy to be discussed in Section 3.3). This means the key encoder is used exactly once per image in the entire process, despite the two appearances in Figure 1 (which is for brevity).

---

[2]That is, matching between two points does not depend on whether they are query or memory points (not true in STM [18] as they are from different encoders).

**Architecture.** Following the STM practice [18], we take `res4` features with stride 16 from the base ResNets as our backbone features and discard `res5`. A $3 \times 3$ convolutional layer without non-linearity is used as a projection head from the backbone feature to either the **key** space ($C^k$ dimensional) or the **value** space ($C^v$ dimensional). We set $C^v$ to be 512 following STM and discuss the choice of $C^k$ in Section 4.1.

**Feature reuse.** As seen from Figure 1, both the key encoder and the value encoder are processing the same frame, albeit with different inputs. It is natural to reuse features from the key encoder (with fewer inputs and a deeper network) at the value encoder. To avoid bloating the feature dimensions and for simplicity, we concatenate the last layer features from both encoders (before the projection head) and process them with two ResBlocks [62] and a CBAM block[3] [64] as the final *value* output.

## 3.2 Memory Reading and Decoding

Given $T$ memory frames and a query frame, the feature extraction step would generate the followings: memory key $\mathbf{k}^M \in \mathbb{R}^{C^k \times THW}$, memory value $\mathbf{v}^M \in \mathbb{R}^{C^v \times THW}$, and query key $\mathbf{k}^Q \in \mathbb{R}^{C^k \times HW}$, where $H$ and $W$ are (stride 16) spatial dimensions. Then, for any similarity measure $c : \mathbb{R}^{C^k} \times \mathbb{R}^{C^k} \rightarrow \mathbb{R}$, we can compute the pairwise affinity matrix $\mathbf{S}$ and the softmax-normalized affinity matrix $\mathbf{W}$, where $\mathbf{S}, \mathbf{W} \in \mathbb{R}^{THW \times HW}$ with:

$$\mathbf{S}_{ij} = c(\mathbf{k}_i^M, \mathbf{k}_j^Q) \qquad \mathbf{W}_{ij} = \frac{\exp{(\mathbf{S}_{ij})}}{\sum_n (\exp{(\mathbf{S}_{nj})})}, \tag{1}$$

where $\mathbf{k}_i$ denotes the feature vector at the $i$-th position. The similarities are normalized by $\sqrt{C^k}$ as in standard practice [18, 33] and is not shown for brevity. In STM [18], the dot product is used as $c$. Memory reading regularization like KMN [22] or top-$k$ filtering [21] can be applied at this step.

With the normalized affinity matrix $\mathbf{W}$, the aggregated readout feature $\mathbf{v}^Q \in \mathbb{R}^{C^v \times HW}$ for the query frame can be computed as a weighted sum of the memory features with an efficient matrix multiplication:

$$\mathbf{v}^Q = \mathbf{v}^M \mathbf{W}, \tag{2}$$

which is then passed to the decoder for mask generation.

In the case of multi-object segmentation, only Equation 2 has to be repeated as $\mathbf{W}$ is defined between image features only, and thus is the same for different objects. In the case of STM [18], $\mathbf{W}$ must be recomputed instead. Detailed running time analysis can be found in Section 6.2.

**Decoder.** Our decoder structure stays close to that of the STM [18] as it is not the focus of this paper. Features are processed and upsampled at a scale of two gradually with higher-resolution features from the key encoder incorporated using skip-connections. The final layer of the decoder produces a stride 4 mask which is bilinearly upsampled to the original resolution. In the case of multiple objects, soft aggregation [18] of the output masks is used.

## 3.3 Memory Management

So far we have assumed the existence of a memory bank of size $T$. Here, we will describe the construction of the memory bank. For each *memory frame*, we store two items: *memory key* and *memory value*. Note that all memory frames (except the first one) are once query frames. The memory key is simply reused from the query key, as described in Section 3.1 without extra computation. The memory value is computed after mask generation of that frame, independently for each object as the value encoder takes both the image and the object mask as inputs.

STM [18] consider every fifth query frame as a memory frame, and the immediately previous frame as a temporary memory frame to ensure accurate matching. In the case of STCN, we find that it is unnecessary, and in fact harmful, to include the last frame as temporary memory. This is a direct consequence of using shared key encoders – 1) key features are sufficiently robust to match well without the need for close-range (temporal) propagation, and 2) the temporary memory key would otherwise be too similar to that of the query, as the image context usually changes smoothly and we do not have the encoding noises resultant from distinct encoders, leading to drifting.[4] This modification also reduces the number of calls to the value encoder, contributing a significant speedup.

---

[3]We find this block to be non-essential in a later experiment but it is kept for consistency.

[4]This effect is amplified by the use of L2 similarity. See the supplementary material for a full comparison.

Table 1: Performance comparison between STM and STCN under different memory configurations on the DAVIS 2017 validation set [65].

| | STM | | STCN | |
|---|---|---|---|---|
| | Every 5$^{\text{th}}$ + Last | Every 5$^{\text{th}}$ only | Every 5$^{\text{th}}$ + Last | Every 5$^{\text{th}}$ only |
| $\mathcal{J}\&\mathcal{F}$ | 82.7 | 81.0 | 83.1 | **85.4** |
| FPS | 12.3 | 16.7 | 15.4 | **20.2** |

Table 1 tabulates the performance comparisons between STM and STCN. For a video of length $L$ with $m \geq 1$ objects, and a final memory bank of size $T < L$, STM [18] would need to invoke the memory encoder and compute the affinity $mL$ times. Our proposed STCN, on the other hand, only invokes the value encoder $mT$ times and computes the affinity $L$ times. It is therefore evident that STCN is significantly faster. Section 6.2 provides a breakdown of running time.

## 4 Computing Affinity

The similarity function $c : \mathbb{R}^{C^k} \times \mathbb{R}^{C^k} \to \mathbb{R}$ plays a crucial role in both STM and STCN, as it supports the construction of affinity that is central to both correspondences and memory reading. It also has to be fast and memory-efficient as there can be up to 50M pairwise relations ($THW \times HW$) to compute for just one query frame.

To recap, we need to compute the similarity between a memory key $\mathbf{k}^M \in \mathbb{R}^{C^k \times HW}$ and a query key $\mathbf{k}^Q \in \mathbb{R}^{C^k \times HW}$. The resultant pairwise affinity matrix is denoted as $\mathbf{S} \in \mathbb{R}^{THW \times HW}$, with $\mathbf{S}_{ij} = c(\mathbf{k}_i^M, \mathbf{k}_j^Q)$ denoting the similarity between $\mathbf{k}_i^M$ (the memory feature vector at the $i$-th position) and $\mathbf{k}_j^Q$ (the query feature vector at the $j$-th position).

In the case of dot product, it can be implemented very efficiently with a matrix multiplication:

$$\mathbf{S}_{ij}^{\text{dot}} = \mathbf{k}_i^M \cdot \mathbf{k}_j^Q \qquad \Rightarrow \qquad \mathbf{S}^{\text{dot}} = \left(\mathbf{k}^M\right)^T \mathbf{k}^Q \tag{3}$$

In the following, we will also discuss the use of cosine similarity and negative squared Euclidean distance as similarity functions. They are defined as (with efficient implementation discussed later):

$$\mathbf{S}_{ij}^{\text{cos}} = \frac{\mathbf{k}_i^M \cdot \mathbf{k}_j^Q}{\left\|\mathbf{k}_i^M\right\|_2 \times \left\|\mathbf{k}_j^Q\right\|_2} \qquad\qquad \mathbf{S}_{ij}^{\text{L2}} = -\left\|\mathbf{k}_i^M - \mathbf{k}_j^Q\right\|_2^2 \tag{4}$$

For brevity, we will use the shorthand "L2" or "L2 similarity" to denote the negative squared Euclidean distance in the rest of the paper. The ranges for dot product, cosine similarity and L2 similarity are $(-\infty, \infty)$, $[-1, 1]$, and $(-\infty, 0]$ respectively. Note that cosine similarity has a limited range. Non-related points are encouraged to have a low similarity score through back-propagation such that they have a close-to-zero affinity (Eq. 1), and thus no value is propagated (Eq. 2).

### 4.1 A Closer Look at the Affinity

The affinity matrix is core to STCN and deserves close attention. Previous works [18, 21, 22, 23, 24], almost by default, use the dot product as the similarity function – *but is this a good choice?*

Cosine similarity computes the angle between two vectors and is often regarded as the normalized dot product. Reversely, we can consider dot product as a scaled version of cosine similarity, with the scale equals to the product of vectors' norms. Note that this is query-agnostic, meaning that every similarity with a memory key $\mathbf{k}_i^M$ will be scaled by its norm. If we cast the aggregation process (Eq. 2) as voting with similarity representing the weights, memory keys with large magnitudes will predominately suppress any representation from other memory nodes.

Figure 2 visualizes this phenomenon in a 2D feature space. For dot product, only a subset of points (labeled as triangles) has a chance to contribute the most for *any query*. Outliers (top-right red) can suppress existing clusters; clusters with dominant value in one dimension (top-left cyan) can suppress other clusters; some points may be able to contribute the most in a region even it is *outside* of the region (bottom-right beige). These undesirable situations will however not happen if the proposed L2 similarity is used: a Voronoi diagram [66] is formed and every memory point can be fully utilized, leading to a *diversified, query-specific* voting mechanism with ease.

Figure 3 shows a closer look at the same problem with soft weights. With dot product, the blue/green point has low weights for every possible query in the first quadrant while a smooth transition is created with our proposed L2 similarity. Note that cosine similarity has the same benefits, but its limited range $[-1, 1]$ means that an extra softmax temperature hyperparameter is required to shape the affinity distribution, or one more parameter to tune. L2 works well without extra temperature tuning in our experiments.

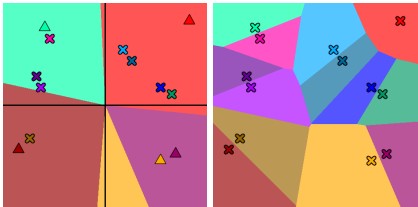

Figure 2: Regions are colored as the "most similar" point under a measure. **Left**: Dot product; **right**: L2 similarity.

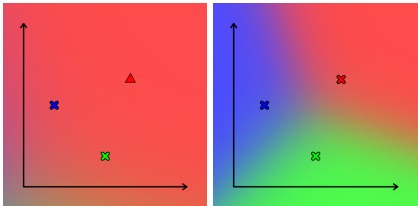

Figure 3: Visualization of the softmax contributions from three points. **Left**: Dot product; **right**: L2 similarity.

**Connection to self-attention, and whether some points are more important than others.** Dot-products have been used extensively in self-attention models [33, 67, 68]. One way to look at the dot-product affinity positively is to consider the points with large magnitudes as *more important* – naturally they should enjoy a higher influence. Admittedly, this is probably true in NLP [33] where a stop word ("the") is almost useless compared to a noun ("London") or in video classification [67] where the foreground human is far more important than a pixel in the plain blue sky. This is however not true for STCN where pixels are more or less equal. It is beneficial to match every pixel in the query frame accurately, including the background (also noted by [10]). After all, if we can know that a pixel is part of the background, we would also know that it does not belong to the foreground. In fact, we find STCN can track the background fairly well (floor, lake, etc.) even when it is never explicitly trained to do so. The notion of relative importance therefore does not generally apply in our context.

**Efficient implementation.** The naïve implementation of negative squared Euclidean distance in Eq. 4 needs to materialize a $C^k \times THW \times HW$ element-wise difference matrix which is then squared and summed. This process is much slower than simple dot product and cannot be run on the same hardware. A simple decomposition greatly simplifies the implementation, as noted in [69]:

$$\mathbf{S}_{ij}^{\text{L2}} = -\left\|\mathbf{k}_i^M - \mathbf{k}_j^Q\right\|_2^2 = 2\mathbf{k}_i^M \cdot \mathbf{k}_j^Q - \left\|\mathbf{k}_i^M\right\|_2^2 - \left\|\mathbf{k}_j^Q\right\|_2^2 \tag{5}$$

which has only slightly more computation than the baseline dot product, and can be implemented with standard matrix operations. In fact, we can further drop the last term as softmax is invariant to translation in the target dimension (details in the supplementary material). For cosine similarity, we first normalize the input vectors, then compute dot product. Table 2 tabulates the actual computational and memory costs.

## 4.2 Experimental Verification

Here, we verify three claims: 1) the aforementioned phenomenon does happen in a high-dimension key space for real-data and a fully-trained model; 2) using L2 similarity diversifies the voting; 3) L2 similarity brings about higher efficiency and performance.

**Affinity distribution.** We verify the first two claims by training two different models with dot product and L2 similarity respectively as the similarity function and plot the maximum contribution given by each memory node in its lifetime. We use the same setting for the two models and report the distribution on the DAVIS 2017 [65] dataset.

Figure 4 shows the pertinent distributions. Under the L2 similarity measure, a lot more memory nodes contribute a fair share. Specifically, around 3% memory nodes never contribute more than 1% weight under dot product while only 0.06% suffer the same fate with L2. Under dot product, 31% memory nodes contribute less than 10% weight at best while the same only happen for 7% of the memory with L2 similarity. To measure the distribution inequality, we additionally compute the Gini coefficient [70] (the higher it is, the more unequal the distribution). The Gini coefficient for dot product is 44.0, while the Gini coefficient for L2 similarity is much lower at 31.8.

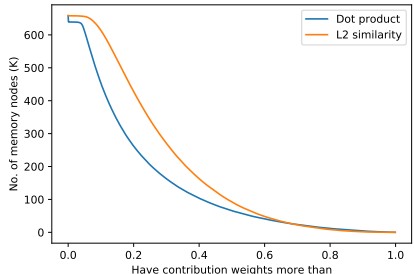

Figure 4: The curves show the number of memory nodes that have contributed above a certain threshold at least once.

**Performance and efficiency.** Next, we show that using L2 similarity does improve performance with negligible overhead. We compare three similarity measures: dot product, cosine similarity, and L2 similarity. For cosine similarity, we use a softmax temperature of 0.01 while a default temperature of 1 is used for both dot product and L2 similarity. This scaling is crucial for cosine similarity only since it is the only one with a limited output range $[-1, 1]$. Searching for an extra hyperparameter is computationally demanding – we simply picked one that converges fairly quickly without collapsing. Table 2 tabulates the main results.

Interestingly, we find that reducing the key space dimension ($C^k$) is beneficial to both cosine similarity and L2 similarity but not dot product. This can be explained in the context of Section 4.1 – the network needs more dimensions so that it can spread the memory key features out to save them from being suppressed by high-magnitude points. Cosine similarity and L2 similarity do not suffer from this problem and can utilize the full key space. The reduced key space in turn benefits memory efficiency and improves running time.

Table 2: Performance comparison between different similarity functions and key space dimensionality ($C^k$) in STCN on the DAVIS 2017 validation set [65]. The number of floating-point operations (FLOPs) are computed for Eq. 3 and Eq. 4 only with $T = 10$. L2 works the best with a reduced key space and a small computational overhead.

| Similarity function | $C^k$ | $\mathcal{J}\&\mathcal{F}$ | # FLOPs (G) | Size of keys (MB) |
|---|---|---|---|---|
| Dot product | 128 | 84.1 | 6.26 | 8.70 |
| Cosine similarity | 128 | 82.6 | 6.26 | 8.70 |
| L2 similarity | 128 | 85.0 | 6.33 | 8.70 |
| Dot product | 64 | 83.2 | **3.13** | **4.35** |
| Cosine similarity | 64 | 83.4 | **3.13** | **4.35** |
| L2 similarity | 64 | **85.4** | 3.20 | **4.35** |

# 5 Implementation Details

Models are trained with two 11GB 2080Ti GPUs with the Adam optimizer [71] using PyTorch [72]. Following previous practices [18, 21], we first pretrain the model on static image datasets [73, 74, 75, 76, 77] with synthetic deformation then perform main training on YouTubeVOS [78] and DAVIS [3, 65]. We also experimented with the synthetic dataset BL30K [79, 80] proposed in [21] which is not used unless otherwise specified. We use a batch size of 16 during pretraining and a batch size of 8 during main training. Pre-training takes about 36 hours and main training takes around 16 hours with batchnorm layers frozen during training following [18]. Bootstrapped cross entropy is used following [21]. The full set of hyperparameters can be found in the open-sourced code.

In each iteration, we pick three temporally ordered frames (with the ground-truth mask for the first frame) from a video to form a training sample [18]. First, we predict the second frame using the first frame as memory. The prediction will be saved as the second memory frame, and then the third frame will be predicted using the union of the first and the second frame. The temporal distance between the frames will first gradually increase from 5 to 25 as a curriculum learning schedule and anneal back to 5 towards the end of training. This process follows the implementation of MiVOS [21].

For memory-read augmentation, we experimented with kernelized memory reading [22] and top-$k$ filtering [21]. We find that top-$k$ works well universally and improves running time while kernelized memory reading is slower and does not always help. We find that $k = 20$ always works better for STCN (original paper uses $k = 50$) and we adopt top-$k$ filtering in all our experiments with $k = 20$. For fairness, we also re-run all experiments in MiVOS [21] with $k = 20$, and pick the best result in their favor. We use L2 similarity with $C^k = 64$ in all experiments unless otherwise specified.

For inference, a 2080Ti GPU is used with full floating point precision for a fair running time comparison. We memorize every 5th frame and no temporary frame is used as discussed in Section 3.3.

## 6 Experiments

We mainly conduct experiments in the **DAVIS 2017 validation** [65] set and the **YouTubeVOS 2018** [78] validation set. For completeness, we also include results in the single object **DAVIS 2016 validation** [3] set and the expanded **YouTubeVOS 2019** [78] validation set. Results for the **DAVIS 2017 test-dev** [65] set are included in the supplementary material. We first conduct quantitative comparisons with previous methods, and then analyze the running time for each component in STCN. For reference, we also present results without pretraining on stataic images. Ablation studies have been included in previous sections (Table 1 and Table 2).

### 6.1 Evaluations

Table 3 tabulates the comparisons of STCN with previous methods in semi-supervised video object segmentation benchmarks. For DAVIS 2017 [65], we compare the standard metrics: region similarity $\mathcal{J}$, contour accuracy $\mathcal{F}$, and their average $\mathcal{J}\&\mathcal{F}$. For YouTubeVOS [78], we report $\mathcal{J}$ and $\mathcal{F}$ for both seen and unseen categories, and the averaged overall score $\mathcal{G}$. For comparing the speed, we compute the *multi-object FPS* that is the total number of output frames divided by the total processing time for the entire DAVIS 2017 [65] validation set. We either copy the FPS directly from papers/project websites, or estimate based on their single object inference FPS (simply labeled as $<^5$). We use 480p resolution videos for both DAVIS and YouTubeVOS. Table 4, 5, 6, and 7 tabulate additional results. For the interactive setting, we replace the propagation module of MiVOS [21] with STCN.

**Visualizations.** Figure 6 visualizes the learned correspondences. Note that our correspondences are general and mask-free, naturally associating every pixel (including background bystanders) even when it is only trained with foreground masks. Figure 7 visualizes our semi-supervised mask propagation results with the last row being a failure case (Section 7).

**Leaderboard results.** Our method is also very competitive on the public VOS challenge leaderboard [78]. Methods on the leaderboard are typically cutting-edge, with engineering extensions like deeper network, multi-scale inference, and model ensemble. They usually represent the highest achievable performance at the time. On the latest YouTubeVOS 2019 validation split [78], our base model (84.2 $\mathcal{G}$) outperforms the previous challenge winner [32] (based on STM [18], 82.0 $\mathcal{G}$) by a large margin. With ensemble and multi-scale testing (details in the supplementary material), our method is ranked first place (86.7 $\mathcal{G}$) at the time of submission on the still active leaderboard.

### 6.2 Running Time Analysis

Here, we analyze the running time of each component in STM and STCN on DAVIS 2017 [65]. For a fair comparison, we use our own implementation of STM, enabled top-$k$ filtering [21], and set $C^k = 64$ for both methods such that all the speed improvements come from the fundamental differences between STM and STCN. Our affinity matching time is lower because we compute a single affinity between raw images while STM [18] compute one for every object. Our value encoder takes much less time than the memory encoder in STM [18] because of our light network, feature reuse, and robust memory bank/management as discussed in Section 3.3.

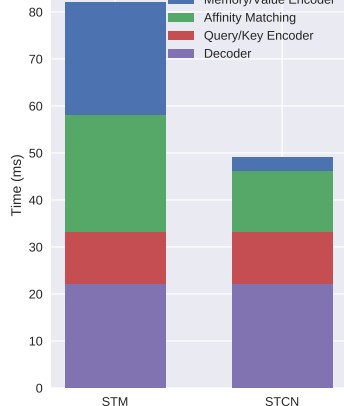

Figure 5: Average running time of each component in STM and STCN.

---

[5]A linear extrapolation would severally underestimate the performance of many previous methods.

Table 3: Comparisons between different methods on DAVIS 2017 and YouTubeVOS 2018 validation sets. Subscripts $S$ and $U$ denote seen or unseen respectively. FPS is measured for multi-object scenarios and is measured on DAVIS 2017. Methods are ranked by YouTubeVOS performance; STM is re-timed on our hardware (see supplementary material); our model is the fastest among methods that are better than STM [18]; * denotes contemporary work.

| Method | YouTubeVOS 2018 [78] | | | | | DAVIS 2017 [65] | | | |
|--------|------|------|------|------|------|------|------|------|------|
| | $\mathcal{G}$ | $\mathcal{J}_S$ | $\mathcal{F}_S$ | $\mathcal{J}_U$ | $\mathcal{F}_U$ | $\mathcal{J\&F}$ | $\mathcal{J}$ | $\mathcal{F}$ | FPS |
| OSMN [8] | 51.2 | 60.0 | 60.1 | 40.6 | 44.0 | 54.8 | 52.5 | 57.1 | <7.1 |
| RGMP [55] | 53.8 | 59.5 | - | 45.2 | - | 66.7 | 64.8 | 68.6 | <7.7 |
| RVOS [12] | 56.8 | 63.6 | 67.2 | 45.5 | 51.0 | 50.3 | 48.0 | 52.6 | <15 |
| Track-Seg [54] | 63.6 | 67.1 | 70.2 | 55.3 | 61.7 | 72.3 | 68.6 | 76.0 | <39 |
| PReMVOS [81] | 66.9 | 71.4 | 75.9 | 56.5 | 63.7 | 77.8 | 73.9 | 81.7 | <0.03 |
| TVOS [82] | 67.8 | 67.1 | 69.4 | 63.0 | 71.6 | 72.3 | 69.9 | 74.7 | 37 |
| FRTM-VOS [6] | 72.1 | 72.3 | 76.2 | 65.9 | 74.1 | 76.7 | - | - | <21.9 |
| GC [24] | 73.2 | 72.6 | 68.9 | 75.6 | 75.7 | 71.4 | 69.3 | 73.5 | <25 |
| SwiftNet* [30] | 77.8 | 77.8 | 81.8 | 72.3 | 79.5 | 81.1 | 78.3 | 83.9 | 25 |
| STM [18] | 79.4 | 79.7 | 84.2 | 72.8 | 80.9 | 81.8 | 79.2 | 84.3 | 10.2 |
| AFB-URR [23] | 79.6 | 78.8 | 83.1 | 74.1 | 82.6 | 74.6 | 73.0 | 76.1 | 4 |
| GraphMem [16] | 80.2 | 80.7 | 85.1 | 74.0 | 80.9 | 82.8 | 80.2 | 85.2 | 5 |
| MiVOS* [21] | 80.4 | 80.0 | 84.6 | 74.8 | 82.4 | 83.3 | 80.6 | 85.9 | 11.2 |
| CFBI [10] | 81.4 | 81.1 | 85.8 | 75.3 | 83.4 | 81.9 | 79.1 | 84.6 | 5.9 |
| KMN [22] | 81.4 | 81.4 | 85.6 | 75.3 | 83.3 | 82.8 | 80.0 | 85.6 | <8.4 |
| RMNet* [29] | 81.5 | 82.1 | 85.7 | 75.7 | 82.4 | 83.5 | 81.0 | 86.0 | <11.9 |
| LWL [7] | 81.5 | 80.4 | 84.9 | 76.4 | 84.4 | 81.6 | 79.1 | 84.1 | <6.0 |
| CFBI+* [83] | 82.0 | 81.2 | 86.0 | 76.2 | 84.6 | 82.9 | 80.1 | 85.7 | 5.6 |
| LCM* [28] | 82.0 | **82.2** | **86.7** | 75.7 | 83.4 | 83.5 | 80.5 | 86.5 | ∼9.2 |
| Ours | **83.0** | 81.9 | 86.5 | **77.9** | **85.7** | **85.4** | **82.2** | **88.6** | **20.2** |
| MiVOS* [21] + BL30K | 82.6 | 81.1 | 85.6 | 77.7 | 86.2 | 84.5 | 81.7 | 87.4 | 11.2 |
| Ours + BL30K | **84.3** | **83.2** | **87.9** | **79.0** | **87.3** | **85.3** | **82.0** | **88.6** | **20.2** |

Figure 6: Visualization of the correspondences. Labels are hand-picked in the source frame (leftmost) and are propagated to the rest directly without intermediate memory. We label all the peaks (e.g., the two yellow diamonds representing the front/back wheel – our algorithm cannot distinguish them). The bystander in white (labeled with an orange crescent) is occluded in the last frame and the resultant affinity does not have a distinct peak (not labeled).

## 7 Limitations

To alienate our method from other possible enhancement, we only use fundamentally simple global matching. Like STM [18], we have no notion of temporal consistency as we do not employ local matching [58, 10, 17] or optical flow [29]. This means we may incorrectly segment objects that are far away with similar appearance. One such failure case is shown on the last row of Figure 7. We expect that given our framework's simplicity, our method can be readily extended to include temporal consistency consideration for further improvement.

## 8 Conclusion

We present STCN, a simple, effective, and efficient framework for video object segmentation. We propose to use direct image-to-image correspondence for efficiency and more robust matching, and examine the inner workings of affinity in details – L2 similarity is proposed as a result of our observations. With its clear technical advantages, We hope that STCN can serve as a new baseline backbone for future contributions.

Table 4: Results on the DAVIS 2016 validation set.

| Method | $\mathcal{J}\&\mathcal{F}$ | $\mathcal{J}$ | $\mathcal{F}$ |
|---|---|---|---|
| OSMN [8] | 73.5 | 74.0 | 72.9 |
| MaskTrack [15] | 77.6 | 79.7 | 75.4 |
| OSVOS [5] | 80.2 | 79.8 | 80.6 |
| FAVOS [14] | 81.0 | 82.4 | 79.5 |
| FEELVOS [58] | 81.7 | 81.1 | 82.2 |
| RGMP [55] | 81.8 | 81.5 | 82.0 |
| Track-Seg [54] | 83.1 | 82.6 | 83.6 |
| FRTM-VOS [6] | 83.5 | - | - |
| CINN [51] | 84.2 | 83.4 | 85.0 |
| OnAVOS [50] | 85.5 | 86.1 | 84.9 |
| PReMVOS [81] | 86.8 | 84.9 | 88.6 |
| GC [24] | 86.8 | 87.6 | 85.7 |
| RMNet [29] | 88.8 | 88.9 | 88.7 |
| STM [18] | 89.3 | 88.7 | 89.9 |
| CFBI [10] | 89.4 | 88.3 | 90.5 |
| CFBI+ [83] | 89.9 | 88.7 | 91.1 |
| MiVOS [21] | 90.0 | 88.9 | 91.1 |
| SwiftNet [30] | 90.4 | 90.5 | 90.3 |
| KMN [22] | 90.5 | 89.5 | 91.5 |
| LCM [28] | 90.7 | **91.4** | 89.9 |
| Ours | **91.6** | 90.8 | **92.5** |
| MiVOS [21] + BL30K | 91.0 | 89.6 | 92.4 |
| Ours + BL30K | **91.7** | 90.4 | **93.0** |

Table 5: Results on the YouTubeVOS 2019 validation set.

| Method | $\mathcal{G}$ | $\mathcal{J}_S$ | $\mathcal{F}_S$ | $\mathcal{J}_U$ | $\mathcal{J}_U$ |
|---|---|---|---|---|---|
| MiVOS [21] | 80.3 | 79.3 | 83.7 | 75.3 | 82.8 |
| CFBI [10] | 81.0 | 80.6 | 85.1 | 75.2 | 83.0 |
| Ours | **82.7** | **81.1** | **85.4** | **78.2** | **85.9** |
| MiVOS [21] + BL30K | 82.4 | 80.6 | 84.7 | 78.1 | 86.4 |
| Ours + BL30K | **84.2** | **82.6** | **87.0** | **79.4** | **87.7** |

Table 6: Results on the DAVIS interactive track [65]. BL30K [21] used for both MiVOS and ours.

| Method | AUC-$\mathcal{J}\&\mathcal{F}$ | $\mathcal{J}\&\mathcal{F}$ @ 60$s$ | Time (s) |
|---|---|---|---|
| ATNet [84] | 80.9 | 82.7 | 55+ |
| STM [85] | 80.3 | 84.8 | 37 |
| GIS [86] | 85.6 | 86.6 | 34 |
| MiVOS [21] | 87.9 | 88.5 | 12 |
| Ours | **88.4** | **88.8** | **7.3** |

Table 7: Effects of pretraining on static images/main-training on the DAVIS 2017 validation set.

| | $\mathcal{J}\&\mathcal{F}$ | $\mathcal{J}$ | $\mathcal{F}$ |
|---|---|---|---|
| Pre-training only | 75.8 | 73.1 | 78.6 |
| Main training only | 82.5 | 79.3 | 85.7 |
| Both | **85.4** | **82.2** | **88.6** |

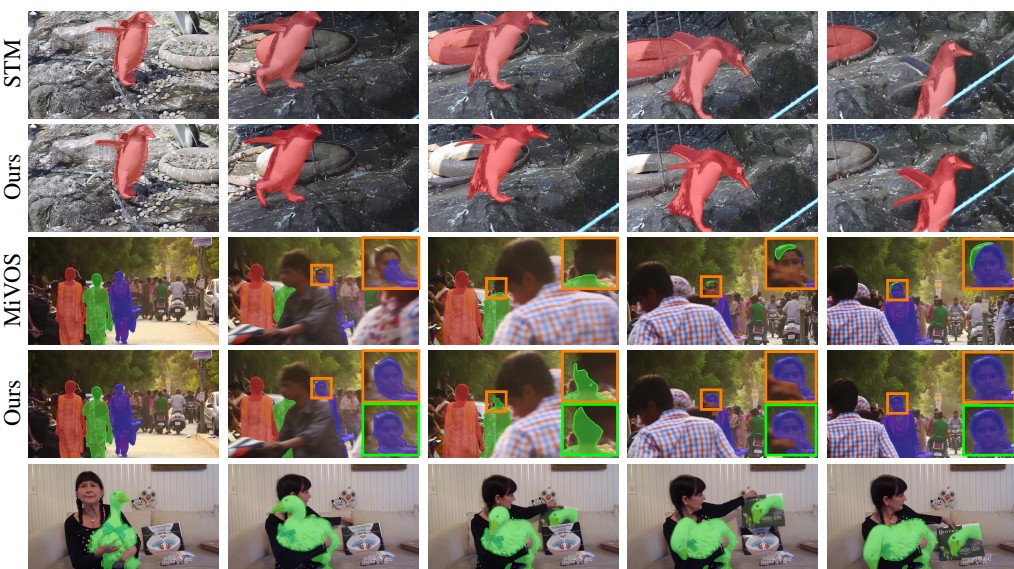

Figure 7: Visualization of semi-supervised VOS results with the first column being the reference masks to be propagated. The first two examples show comparisons of our method with STM [18] and MiVOS [21]. In the second example, zoom-ins inset (orange) are shown with the corresponding ground-truths inset (green) to highlight their differences. The last row shows a failure case: we cannot distinguish the real duck from the duck picture, as no temporal consistency clue is used in our method.

**Broader Impacts** Malicious use of VOS software can bring potential negative societal impacts, including but not limited to unauthorized mass surveillance or privacy infringing human/vehicle tracking. We believe that the task itself is neutral with positive uses as well, such as video editing for amateurs or making safe self-driving cars.

## Acknowledgment

This research is supported in part by Kuaishou Technology and the Research Grant Council of the Hong Kong SAR under grant no. 16201818.

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
