# Supplementary Material for STCN

## A    PyTorch-style Pseudocode

We first prove that the $\left\|\mathbf{k}_j^Q\right\|_2^2$ term will be canceled out in the subsequent softmax operation:

$$
\begin{aligned}
\mathbf{W}_{ij} &= \frac{\exp\left(\mathbf{S}^{\mathrm{L2}}{}_{ij}\right)}{\sum_n \exp\left(\mathbf{S}_{nj}^{\mathrm{L2}}\right)} \\
&= \frac{\exp\left(2\mathbf{k}_i^M \cdot \mathbf{k}_j^Q - \left\|\mathbf{k}_i^M\right\|_2^2 - \left\|\mathbf{k}_j^Q\right\|_2^2\right)}{\sum_n \exp\left(2\mathbf{k}_n^M \cdot \mathbf{k}_j^Q - \|\mathbf{k}_n^M\|_2^2 - \left\|\mathbf{k}_j^Q\right\|_2^2\right)} \\
&= \frac{\exp\left(2\mathbf{k}_i^M \cdot \mathbf{k}_j^Q - \left\|\mathbf{k}_i^M\right\|_2^2\right) / \exp\left(\left\|\mathbf{k}_j^Q\right\|_2^2\right)}{\sum_n \exp\left(2\mathbf{k}_n^M \cdot \mathbf{k}_j^Q - \|\mathbf{k}_n^M\|_2^2\right) / \exp\left(\left\|\mathbf{k}_j^Q\right\|_2^2\right)} \\
&= \frac{\exp\left(2\mathbf{k}_i^M \cdot \mathbf{k}_j^Q - \left\|\mathbf{k}_i^M\right\|_2^2\right)}{\sum_n \exp\left(2\mathbf{k}_n^M \cdot \mathbf{k}_j^Q - \|\mathbf{k}_n^M\|_2^2\right)}
\end{aligned}
$$

Then, we show our efficient PyTorch [1] style implementation of Eq. 5.

---

**Algorithm 1:** L2 Similarity Computation

```
// Input:
//    k^M : C^k × THW
//    k^Q : C^k × HW
// Returns:
//    S : THW × HW
1 function GetSimilarity(k^M, k^Q)
    // Decomposing L2 distance into two parts:  ab, a_sq
    // b_sq is canceled out as shown above.
2   ab ← k^M.transpose().matmul(k^Q) // ab : THW × HW
3   a_sq ← k^M.pow(2).sum(dim=0).unsqueeze(dim=1) // a_sq : THW × 1
    // This step utilizes broadcasting:  dimensions of size 1 will be
       implicitly expanded to match other tensors without extra memory
       cost
4   S ← 2 * ab − a_sq // S : THW × HW
5   return S
```

---

## B    Additional results on memory scheduling

Table 1 tabulates additional quantitative results of STM/STCN with dot product/L2 similarity under different memory scheduling strategies.

1. With STM, there is a significant performance drop when the temporary frame is not used regardless of the similarity function. This supports our claim that STM requires close-range relations for robust matching.

2. With STCN+dot product, the performance drop is slight. Note that by dropping the temporary frame, we also enjoy a 31% increase in FPS as shown in the main paper. This supports that the STCN architecture provides more robust key features.

3. With STCN+L2 similarity, it performs the best when the temporary frame is not used (while also enjoying the 31% speed improvement). This suggests L2 similarity is more susceptible to drifting when the memory scheduling is not carefully designed. Overall, L2 similarity is still beneficial.

Table 1: Performance ($\mathcal{J}\&\mathcal{F}$) of networks under different memory configurations on the DAVIS 2017 validation set [2].

| | Every 5$^{th}$ + Last | Every 5$^{th}$ only |
|---|---|---|
| STM + Dot product | 83.3 | 81.3 |
| STM + L2 similarity | 82.7 | 81.0 |
| STCN + Dot product | 84.3 | 84.1 |
| STCN + L2 similarity | 83.1 | 85.4 |

## C  DAVIS `test-dev`

The `test-dev` split of DAVIS 2017 is notably more difficult than the training or validation set with rapid illumination changes and highly similar objects. As a result, methods with strong spatial constraints like KMN [3] or CFBI [4] typically perform better (while still suffering a drop in performance from validation to `test-dev`). Some would use additional engineering, such as using 600p videos instead of the standard 480p [5, 3, 6].

Our baseline is severally punished for lacking spatial cues. BL30K helps a lot in this case but is still not enough to reach SOTA performance. We find that simply encoding memory more frequently, i.e., every third frame instead of every fifth frame, can boost the performance of the model with extended training to be above state-of-the-art on DAVIS `test-dev`. Admittedly, this comes with additional computational and memory costs. Thanks to our efficient base model, STCN with increased memory frequency is still runnable on the same 11GB GPU and is the fastest among competitors. Table 2 tabulates the comparison with other methods. FPS for methods that use 600p evaluation is estimated from that of 480p (assuming linear scaling) unless given in their original paper.

Table 2: Comparisons between competitive methods on DAVIS 2017 `test-dev` [2]. M3 denotes increased memory frequency. ∗ denotes 600p evaluation and † denotes the use of spatial cues – our baseline is severally punished for lacking it which is crucial in this particular data split. Subscript arrow denotes changes from baseline. Some results are modified from the table in [6]. [1]

| Method | DAVIS 2017 `test-dev` [2] | | | |
|---|---|---|---|---|
| | $\mathcal{J}\&\mathcal{F}$ | $\mathcal{J}$ | $\mathcal{F}$ | FPS |
| OSMN [7] | 41.3 | 37.3 | 44.9 | <2.38 |
| OnAVOS [8] | 56.5 | 53.4 | 59.6 | <0.03 |
| RGMP [9] | 52.9 | 51.3 | 54.4 | <2.38 |
| FEELVOS [10] | 57.8 | 55.2 | 60.5 | <1.85 |
| PReMVOS [11] | 71.6 | 67.5 | 75.7 | <0.02 |
| STM ∗ [5] | 72.2 | 69.3 | 75.2 | 6.5 |
| RMNet † [12] | 75.0 | 71.9 | 78.1 | <11.9 |
| CFBI ∗† [4] | 76.6 | 73.0 | 80.1 | 2.9 |
| KMN ∗† [3] | 77.2 | 74.1 | 80.3 | <5.4 |
| CFBI+ ∗† [6] | 78.0 | 74.4 | 81.6 | 3.4 |
| LCM † [13] | 78.1 | 74.4 | 81.8 | 9.2 |
| MiVOS† [14] + BL30K | 78.6 | 74.9 | 82.2 | 10.7 |
| Ours | 76.1 | 72.7 | 79.6 | **20.2** |
| Ours, M3 | 76.5$_{\uparrow 0.4}$ | 73.1$_{\uparrow 0.4}$ | 80.0$_{\uparrow 0.4}$ | 14.6$_{\downarrow 5.6}$ |
| Ours + BL30K | 77.8$_{\uparrow 1.7}$ | 74.3$_{\uparrow 1.6}$ | 81.3$_{\uparrow 1.7}$ | **20.2** |
| Ours + BL30K, M3 | **79.9**$_{\uparrow 3.8}$ | **76.3**$_{\uparrow 3.6}$ | **83.5**$_{\uparrow 3.9}$ | 14.6$_{\downarrow 5.6}$ |

## D  Re-timing on Our Hardware

The inference speed of networks (even with the same architecture) can be influenced by implementations, software package versions, or simply hardware. As we are mainly evolving STM [5] into STCN, we believe that it is just to re-time our re-implemented STM with our software and hardware

for a fair comparison. While the original paper reported 6.25 single-object FPS (which will only be slower for multi-object), we obtain 10.2 multi-object FPS which is reported in our Table 3. We believe this gives the proper acknowledgment that STM deserves.

## E   Multi-scale Testing and Ensemble

To compete with other methods that use multi-scale testing and/or model ensemble on the YouTubeVOS 2019 validation set [15], we also try these techniques on our model. Table 3 tabulates the results. Our base model beats the previous winner [16], and our ensemble model achieves top-1 on the still active validation leaderboard at the time of writing. The 2021 challenge has not released the official results/technical reports by NeurIPS 2021 deadline, but our method outperforms all of them on the validation set.

For multi-scale testing, we feed the network with 480p/600p and original/flipped images and simply average the output probabilities. As we trained our network with 480p images, we additionally employ kernelized memory reading [3] in 600p inference for a stronger regularization.

For model ensemble, we adopt three model variations: 1) original model trained with BL30K [14], 2) backbone replaced by WideResNet-50 [17], and 3) backbone replaced by WideResNet-50 [17] + ASPP [18]. Their outputs are simply averaged as in the case for multi-scale inference. We do not try deeper models due to computational constraints.

Table 3: Results on the YouTubeVOS 2019 validation split [15]. MS denotes multi-scale testing. EMN [16] is the previous challenge winner. Methods are ranked by performance, with variants of the same method grouped together.

| Method | MS? | Ensemble? | $\mathcal{G}$ | $\mathcal{J}_S$ | $\mathcal{F}_S$ | $\mathcal{J}_U$ | $\mathcal{J}_U$ |
|---|---|---|---|---|---|---|---|
| EMN [16] | ✓ | ✓ | 82.0 | - | - | - | - |
| CFBI [4] | ✗ | ✗ | 81.0 | 80.6 | 85.1 | 75.2 | 83.0 |
| CFBI [4] | ✓ | ✗ | 82.4 | 81.8 | 86.1 | 76.9 | 84.8 |
| MiVOS [14] | ✗ | ✗ | 82.4 | 80.6 | 84.7 | 78.1 | 86.4 |
| Ours | ✗ | ✗ | 84.2 | 82.6 | 87.0 | 79.4 | 87.7 |
| Ours | ✓ | ✗ | 85.2 | 83.5 | 87.8 | 80.7 | 88.7 |
| Ours | ✓ | ✓ | **86.7** | **85.1** | **89.6** | **82.2** | **90.0** |

## F   Implementation Details

### F.1   Optimizer

The Adam [19] optimizer is used with default momentum $\beta_1 = 0.9, \beta_2 = 0.999$, a base learning rate of $10^{-5}$, and a L2 weight decay of $10^{-7}$. During training, we use PyTorch's [1] Automatic Mixed Precision (AMP) for speed up with automatic gradient scaling to prevent `NaN`. It is not used during inference.

Batchnorm layers are set to `eval` mode during training, meaning that their running mean and variance are not updated and batch-wise statistics are not computed to save computational and memory cost following STM [5]. The normalization parameters are thus kept the same as the initial ImageNet pretrained network. The affine parameters $\gamma$ and $\beta$ are still updated through back-propagation.

### F.2   Feature Fusion

As mentioned in the main text (feature reuse), we fuse the last layer features from both encoders (before the projection head) with two ResBlocks [20] and a CBAM block [21] as the final *value* output. To be more specific, we first concatenate the features from the key encoder (1024 channels) and the features from the value encoder (256 channels) and pass them through a `BasicBlock` style

---

[1]MiVOS [14] uses spatial cues from kernelized memory reading [3] *in and only in* their DAVIS `test-dev` evaluation.

ResBlock with two $3 \times 3$ convolutions which output a tensor with 512 channels. Batchnorm is not used. We maintain the channel size in the rest of the fusion block, and pass the tensor through a CBAM block [21] under the default setting and another `BasicBlock` style ResBlock.

### F.3 Dataset

**All the data augmentation strategies follow exactly from the open-sourced training code of MiVOS [14] to avoid engineering distractions**. They are mentioned here for completeness, but readers are encouraged to look at the implementation directly.

#### F.3.1 Pre-training

We used ECSSD [22], FSS1000 [23], HRSOD [24], BIG [25], and both the training and testing set of DUTS [26]. We downsized BIG and HRSOD images such that the longer edge has 512 pixels. The annotations in BIG and HRSOD are of higher quality and thus they appear five times more often than images in other datasets. We use a batch size of 16 during the static image pretraining stage with each data sample containing three synthetic frames augmented from the same image.

For augmentation, we first perform PyTorch's random scaling of $[0.8, 1.5]$, random horizontal flip, random color jitter of (`brightness=0.1, contrast=0.05, saturation=0.05, hue=0.05`), and random grayscale with a probability of 0.05 on the base image. Then, for each synthetic frame, we perform PyTorch's random affine transform with rotation between $[-20, 20]$ degrees, scaling between $[0.9, 1.1]$, shearing between $[-10, 10]$ degrees, and another color jitter of (`brightness=0.1, contrast=0.05, saturation=0.05`). The output image is resized such that the shorter side has a length of 384 pixels, and then a random crop of 384 pixels is taken. With a probability of $33\%$, the frame undergoes an additional thin-plate spline transform [27] with 12 randomly selected control points whose displacements are drawn from a normal distribution with scale equals 0.02 times the image dimension (384).

#### F.3.2 Main training

We use the training set of YouTubeVOS [15] and DAVIS [2] 480p in main training. All the images in YouTubeVOS are downsized such that shorter edge has 480 pixels, like those in the DAVIS 480p set. Annotations in DAVIS has a higher quality and they appear 5 times more often than videos in YouTubeVOS, following STM [5]. We use a batch size of 8 during main training with each data sample containing three temporally ordered frames from the same video.

For augmentation, we first perform PyTorch's random horizontal flip, random resized crop with (`crop_size=384, scale=(0.36, 1), ratio=(0.75, 1.25)`), color jitter of (`brightness=0.1, contrast=0.03, saturation=0.03`), and random grayscale with a probability of 0.05 for every image in the sequence with the same random seed such that every image undergoes the same initial transformation. Then, for every image (with different random seed), we apply another PyTorch's color jitter of (`brightness=0.01, contrast=0.01, saturation=0.01`), random affine transformation with rotation between $[-15, 15]$ degrees and shearing between $[-10, 10]$ degrees. Finally, we pick at most two objects that appear on the first frame as target objects to be segmented.

#### F.3.3 BL30K training

BL30K [14] is a synthetic VOS dataset that can be used after static image pretraining and before main training. It allows the model to learn complex occlusion patterns that do not exist in static images (and the main training datasets are not large enough). The sampling and augmentation strategy is the same as the ones in main training, except 1) cropping scale in the first random resized crop is $(0.25, 1.0)$ instead of $(0.36, 1.0)$ – this is because BL30K images are slightly larger and 2) we discard objects that are tiny in the first frame (<100 pixels) as they are very difficult to learn and unavoidable in these synthetic data.

### F.4 Training iteration and scheduling

By default, we first train on static images for 300K iterations then perform main training for 150K iterations. When we use BL30K [14], we train for 500K iterations on it after static image pretraining

and before main training. Main training would be extended to 300K iterations to combat the domain shift caused by BL30K. Regular training without BL30K takes around 30 hours and extended training with BL30K takes around 4 days in total.

With a base learning rate of $10^{-5}$, we apply step learning rate decay with a decay ratio of $\gamma = 0.1$. In static image pretraining, we perform decay once after 150K iterations. In standard (150K) main training, once after 125K iterations; in BL30K training, once after 450K iterations; in extended (300K) main training, once after 250K iterations. The learning rate scheduling is independent of the training stages, i.e., all training stages start with the base learning rate of $10^{-5}$ regardless of any pretraining.

As for curriculum learning mentioned in Section 5, we apply the same schedule as MiVOS [14]. The maximum temporal distance between frames is set to be $[5, 10, 15, 20, 25, 5]$ at the corresponding iterations of $[0\%, 10\%, 20\%, 30\%, 40\%, 90\%]$ or $[0\%, 10\%, 20\%, 30\%, 40\%, 80\%]$ of the total training iterations for main training and BL30K respectively. A longer annealing time is used for BL30K because it is relatively harder.

### F.5  Loss function

We use bootstrapped cross entropy (or hard pixel-mining) as in MiVOS [14]. Again, the parameters are the same as in MiVOS [14] as we do not aim to over-engineer and are included for completeness.

For the first 20K iterations, we use standard cross entropy loss. After that, we pick the top-$p\%$ pixels that have the highest loss and only compute gradients for these pixels. $p$ is linearly decreased from 100 to 15 over 50K iterations and is fixed at 15 afterward.