# OpenReview forum: "Rethinking Space-Time Networks with Improved Memory Coverage for Efficient Video Object Segmentation"
_NeurIPS.cc/2021/Conference — NeurIPS 2021 Poster_

### Official Review · Reviewer_W1R6 · 2021-07-08

**Rating:** 6
**Confidence:** 5

**Summary:**

The authors propose STCN for video object segmentation. Compared to STM, the dot product similarity is replaced with the L2 similarity. Moreover, the key/value encoders are separated and thus the features can be reused. Though simple, these small changes make STCN archive the STOA results and faster inference speed. Moreover, it gives a closer look at the similarity measure.

**Limitations And Societal Impact:**

1. Though simple, the technical contributions are limited.
2. According to Eq.5, the L2 similarity can be regarded as the dot product similarity (K_i^M \dot K_j^Q) with L2 normalization (K_i^M * K_i^M; K_j^Q * K_j^Q). However, the segmentation accuracy is much higher than the dot product. Why?

**Main Review:**

1. The paper is clearly written.
2. Section 4 gives a good analysis of different similarity measures.
3. The proposed STCN archives the STOA results on mainstream benchmarks.

**Time Spent Reviewing:**

4

---

> ### Author Response · Authors · 2021-08-09
> **Response to Reviewer W1R6**
>
> Thank you for the review and questions. We will address the questions one by one:
>
> **1\. Technical contributions.** We would like to restate that simplicity is one of our goals, and our careful redesign of core modules in the popular STM framework is instrumental to future works in this direction. With our to-be-released open-source code, we believe the new and deep insight with our solid technical STCN contribution and exhaustive experimentation should be of sufficient impact to NeurIPS and future VOS research.
>
> **2\. Dot product with L2 normalization.** True. From an optimization perspective, L2 similarity would be the same as dot product when L2 normalization is enabled (with the normalization discarding one degree of freedom). However, L2 normalization is not trivially learnable via gradient descent as demonstrated by the empirical gap in performance.

---

### Official Review · Reviewer_BydX · 2021-07-11

**Rating:** 7
**Confidence:** 4

**Summary:**

This paper simplifies space-time correspondence with shared Siamese key encoders for all raw frames and gets rid of the unnecessary re-encoding procedure for every object in the same frame. The authors further discuss the similarity functions used in affinity computation and provide comprehensive experiments on memory composition and running time.

**Limitations And Societal Impact:**

Yes, the authors have addressed the limitations and potential negative societal impact of their work .

**Main Review:**

## Pros:
- I enjoy reading the manuscript, which is well-structured. The ideas are well-presented and easy to follow. The proposed pipeline is clear and reasonable albeit with some minor contradictions in description.
- I appreciate the analysis and experiments on similarity functions and running time, which help a lot for better understanding the proposed method and potential issues in previous works.
- The authors promise to release the code, which would be truly valuable.

## Cons:
- About the counterintuitive result on Memory management, where the short-term consistency seems to be harmful to STCN when applying L2 similarity, I wonder what would happen if cosine similarity and dot product similarity are adopted instead of L2. Additionally, I recommend an ablation on merely maintaining the last frame in memory and provide corresponding qualitative results to prove the drifting assumption in Ln152.
- I appreciate the assumption that ‘every pixel counts’ and consent the extra robustness from additional ‘meaningful’ memory nodes. But I think the proposed method does not explicitly explore the correspondence between background pixels, as [9] did, but just spread the ‘attention’ to more foreground pixels, and thus introduces the mismatching issues as described in the limitation section, which makes the discussion in Ln206-215 not that convincing where the authors claim they could benefit from distinguishing the backgrounds. The authors also analyze the different undergoing situations between STCN and other NLP methods or video classification methods, but I could not get the point and the discussion seems a little bit redundant. I would recommend a refinement or some detailed explanation.
+ [9]: Collaborative video object segmentation by foreground-background integration. In ECCV2020.


**Time Spent Reviewing:**

3

---

> ### Author Response · Authors · 2021-08-09
> **Response to Reviewer BydX**
>
> Thank you for the detailed review and questions. We will address the questions one by one:
>
> **1. Results on memory management.** We find that the change of optimal memory scheduling is related to both STCN and L2 similarity (full results tabulated below). We will add relevant discussion in the main paper.
>
> &nbsp;&nbsp;&nbsp;&nbsp;&nbsp;&nbsp; **1.1\.** With STM, there is a significant performance drop when the temporary frame is not used regardless of the similarity function. This supports our claim that STM requires close-range relations for robust matching
>
> &nbsp;&nbsp;&nbsp;&nbsp;&nbsp;&nbsp; **1.2\.** With STCN+dot product, the performance drop is slight. Note that by dropping the temporary frame, we also enjoy a 31% increase in FPS as shown in the main paper. This supports that the STCN architecture provides more robust key features.
>
> &nbsp;&nbsp;&nbsp;&nbsp;&nbsp;&nbsp; **1.3\.** With STCN+L2 similarity, it performs the best when the temporary frame is not used (while also enjoying the 31% speed improvement). This suggests L2 similarity is more susceptible to drifting (L150) when the memory scheduling is not carefully designed. Overall, L2 similarity is still beneficial.
>
> *Performance of networks under different memory scheduling strategies:*
>
> |                      | Every 5th + last frame | Every 5th only |
> |----------------------|:----------------------:|:--------------:|
> | STM + Dot product    |          83.3          |      81.3      |
> | STM + L2 similarity  |          82.7          |      81.0      |
> | STCN + Dot product   |          84.3          |      84.1      |
> | STCN + L2 similarity |          83.1          |      85.4      |
>
> **2. Results with only the last frame as memory.** We would expect the model to perform a lot worse as it cannot adapt to object disappearance or utilize rich features in a large memory bank. We do notice significant drifting in this case and our performance would drop to 73.4.
>
> **3. The background matching explanation.** Thank you for the question. We will clarify in the main paper with the following ideas:
>
> &nbsp;&nbsp;&nbsp;&nbsp;&nbsp;&nbsp; **3.1\.** Unlike explicit foreground/background separation as in [9], we never distinguish foreground/background pixels in the correspondence stage -- we aim to match all of them, regardless of their semantic identities. Explicit separation would unavoidably create biases for foreground/background, in contrast to our holistic approach.
>
> &nbsp;&nbsp;&nbsp;&nbsp;&nbsp;&nbsp; **3.2\.** We believe STCN does not spread attention to foreground pixels any more than STM does for two reasons: 1) STCN focuses on a more diverse use of the memory (which is beneficial according to our experiments), not of the query (which might spread the attention as you suggested), and 2) STM also fails in most of the failure cases that we have presented.
>
> &nbsp;&nbsp;&nbsp;&nbsp;&nbsp;&nbsp; **3.3\.** We perform a simple experiment to perform mask tracking on the background (floor, lake, etc.) and find that the results are pretty accurate -- suggesting that STCN has the ability to perform background matching.

---

### Official Review · Reviewer_NCEc · 2021-07-13

**Rating:** 6
**Confidence:** 5

**Summary:**

This paper presents the Space-Time Correspondence Network (STCN) to tackle the semi-supervised video object segmentation. STCN improves previous STM by building affinity based on RGB-only input and reused key features. Moreover, the author further investigated the computation of similarity and find that negative squared Euclidean distance leads to better performance. Experiments on multiple datasets are conducted to demonstrate the effectiveness of the proposed method.

**Limitations And Societal Impact:**

Don't find limitations and potential negative societal impact of this work.

**Main Review:**

Pros:
+ This paper is clear and easy to read.
+ The proposed method achieves state-of-the-art accuracy and high efficiency.
+ The proposed STCN is reasonable and effective.
+ The investigation about the similarity computation is meaningful.

Cons:
- There may need more intuition for why L2 similarity being better than dot-product similarity. Even though dot-product provides less smooth similarity, it can be relived by the Softmax function. Sec3.1 tries to provide explanations for this, yet the visualization is based on toy examples. It would be better to visualize with real examples.

- In Fig.4, the numbers of total nodes for Dot product and L2 similarity are different. Shouldn't they be the same?

- Are the results in Tab.1 both based on L2 similarity? If not, this should be clarified. Also, it would be better to report performance for STM with L2 similarity.

- I noticed that all the experiments are conducted on the validation sets. It would be better to incorporate results on testing sets into the paper.  From the Tab-1 supplementary, I noticed that the ''ours''  model actually performs worse than the previous method on the test-dev of DAVIS17. This is contradictory to the performance on the validation set.  Any explanation?

- Some related works for fast VOS are missing [a,b].

[a] Fast video object segmentation via dynamic targeting network, cvpr19

[b] Motion-guided cascaded refinement network for video object segmentation,cvpr18


In general, I like this work for its motivation and effective/efficient performance. I would like to see the authors' responses.

**Time Spent Reviewing:**

3

---

> ### Author Response · Authors · 2021-08-09
> **Response to Reviewer NCEc**
>
> Thank you for the detailed review and questions. We will address the questions one by one:
>
> **1\. Intuition for L2, and dot-product with softmax.** Thank you for the suggestion. We would like to emphasize that softmax will tend to *amplify* the uneven similarities instead of smoothing them due to the exponentiation (except when even the most dominant affinities are very small in magnitude --  but that almost never happens in STM/STCN). For example, initial similarities of [1, 4, 1] (max is 4 times the min) will become [0.045, 0.909, 0.045] (max is 20 times the min) after softmax. A similar phenomenon will occur when the affinities are negative.
>
> **2\. Real examples.** We use the 2D illustrative example to give a clean and easily understood visualization. Real data would be too high-dimensional (64 dimensions), with projection likely discarding too much information for the purpose of demonstration. On the other hand, we actually have provided statistical analysis of **real data** in Figure 4 and in the corresponding paragraph (L231-L242).
>
> **3\. Number of nodes in Figure 4.** The number of nodes are in fact the same. The plot for dot product drops quickly (those representing totally unused nodes) which is explained in the left paragraph (L231-L242).
>
> **4\. Results in Table 1.** Both STCN and STM reported in Table 1 use L2 similarity in the spirit of control experiment, changing one variable at a time. Using dot product for STM would not change our conclusion in Table 1 (details given in the table below).
>
> |                      | Every 5th + last frame | Every 5th only |
> |----------------------|:----------------------:|:--------------:|
> | STM + Dot product    |          83.3          |      81.3      |
> | STCN + Dot product (shown in Table 1)  |          84.3          |      84.1      |
> | STCN + L2 similarity (shown in Table 1) |          83.1          |      85.4      |
>
> **5\. Val vs. test set.** Note that the YouTubeVOS validation set is both large-scale (474 videos) and closed-sourced (we have **no access to the ground-truth**), and is thus a trustworthy benchmark on which we have achieved excellent results. The DAVIS test-dev set, as mentioned on L27-L32 of the supplementary material, is much more challenging with cases like harsh lighting conditions and highly similar objects. Methods that employ hand-crafted locality control (which we do not) would be more robust to this unforeseen distribution change. STM also suffers a large performance drop from validation to test-dev (81.8->72.2). In spite of that, Our method already performs competitively without changes, and performs the best with only a small change while still running pretty fast.
>
> **6\. Missing citation.** Thanks. We will add them to the main paper.

---

### Official Review · Reviewer_LxK4 · 2021-07-16

**Rating:** 6
**Confidence:** 5

**Summary:**

This work proposes a space-time correspondence network which improves performance and speed over the standard STM network by finding pixel-level correspondences in individual frames rather than frames+segmentations. Furthermore, this work shows that the dot-product, which is commonly used in STM is suboptimal and they propose an efficient L2 similarity computation which leads to improved correspondences throughout a video. It is evaluated on two standard VOS benchmarks - Youtube-VOS and DAVIS-2017 - and there are extensive analyses and ablations which give insights into the proposed method.

**Limitations And Societal Impact:**

The authors include a section on the limitations of the approach as well as broader impacts.

**Main Review:**

Overall, the paper is well written and there are no major issues. The proposed approach seems novel, and achieves strong VOS performance when compared with previous SOTA methods. Furthermore, the major claims made by the work are supported by experiments or ablations. There are some questions/issues (see below), so I suggest this work is *marginally above the acceptance threshold* until these are addressed.

## Questions/Issues

1. A recent paper [1] has a similar network architecture, where the correspondence/affinity between pixels is computed separate from the value/query. Although the specific problems on which the networks are applied are different - [1] attempts to perform VOS while training in an unsupervised manner, whereas this work perform the traditional semi-supervised setting - can the authors comment  on the differences between the two approaches? Some discussion differentiating the two methods/architectures would help highlight the novelty of this approach.

2. Table 2 shows that L2 leads to better results on STCN than dot-product and cosine similarity. Is this, however, specific to the STCN, or would replacing the dot-product in another network, e.g. STM, lead to a similar performance improvement?

3. Table 1 uses dot-product for STM and L2 Similarity for the STCN. Would the drop in performance when using the last frames (results in col 3 and col 4) be found when using the dot-product with STCN? Is this specific to the STCN or the L2 Similarity? An ablation exploring this would be beneficial to better understand the strengths/weaknesses of the proposed approach.

## Minor Questions/Comments

1. Is the use of Resnet+CBAM layers to combine query+values (line 117) necessary for strong performance?

2. How would the light value encoder differ from resizing the binary segmentation mask and using the resized mask as $v^M$?

3. It would be useful to see scores when not using static image pretraining, so that the method could be compared with other methods that do not use such data (i.e. only training on Youtube-VOS).

[1] Lai, Z., Lu, E., & Xie, W. (2020). MAST: A memory-augmented self-supervised tracker. In Proceedings of the IEEE/CVF Conference on Computer Vision and Pattern Recognition (pp. 6479-6488).

**Time Spent Reviewing:**

6

---

> ### Author Response · Authors · 2021-08-09
> **Response to Reviewer LxK4**
>
> Thank you for the detailed review and questions. We will address the questions one by one:
>
> **1\.** **Comparison with MAST.** Our network is different from MAST in two important aspects: motivation (which comes from the fact that we are doing different tasks as you mentioned) and the use of a value encoder.
>
> &nbsp;&nbsp;&nbsp;&nbsp;&nbsp;&nbsp; **1.1\.** For self-supervised learning as in MAST, they only utilize RGB images as they do not have access to other inputs. Hence an encoder that takes only RGB is a necessity. In STCN, we deliberately take away the mask input with clearly motivated rationales (L99-L103), and establish empirically that it is a better choice instead of a concession.
>
> &nbsp;&nbsp;&nbsp;&nbsp;&nbsp;&nbsp; **1.2\.** We utilize a value encoder to generate deep features for the mask, while MAST simply downsamples the mask in test-time. Our method makes it more robust to accumulation error, as object features are re-encoded every frame instead of just linearly combining previous masks. With more expressive deep features, we can also use a larger image stride in matching (we use 16, while MAST uses 4) -- our full affinity matrix would therefore be $4^4=256$ times smaller than that of MAST (neglecting the memory-saving engineering extension in MAST that we could also employ in principle).
>
> &nbsp;&nbsp;&nbsp;&nbsp;&nbsp;&nbsp; **1.3\.** We will add relevant discussion in the paper.
>
> **2\.** **STM with L2 similarity.** Thank you for this interesting question. STCN and L2 similarity should be taken holistically -- our rationale and intuition for L2 similarity in Section 4 are based on STCN. Notably, the memory/query keys are in the same embedding space in STCN while not in STM. We find that STM performs slightly worse when L2 similarity is used instead of dot product (83.3 -> 82.7 J&F), with dot product using twice the number of channels to perform well as suggested by Table 2. We will add this observation in the paper.
>
> **3\.** **Memory scheduling with dot product.** We find that the change of optimal memory scheduling is related to both STCN and L2 similarity (full results tabulated below). We will add relevant discussion in the main paper.
>
> &nbsp;&nbsp;&nbsp;&nbsp;&nbsp;&nbsp; **3.1\.** With STM, there is a significant performance drop when the temporary frame is not used regardless of the similarity function. This supports our claim that STM requires close-range relations for robust matching
>
> &nbsp;&nbsp;&nbsp;&nbsp;&nbsp;&nbsp; **3.2\.** With STCN+dot product, the performance drop is slight. Note that by dropping the temporary frame, we also enjoy a 31% increase in FPS as shown in the main paper. This supports that the STCN architecture provides more robust key features.
>
> &nbsp;&nbsp;&nbsp;&nbsp;&nbsp;&nbsp; **3.3\.** With STCN+L2 similarity, it performs the best when the temporary frame is not used (while also enjoying the 31% speed improvement). This suggests L2 similarity is more susceptible to drifting (L150) when the memory scheduling is not carefully designed. Overall, L2 similarity is still beneficial.
>
> *Performance of networks under different memory scheduling strategies:*
>
> |                      | Every 5th + last frame | Every 5th only |
> |----------------------|:----------------------:|:--------------:|
> | STM + Dot product    |          83.3          |      81.3      |
> | STM + L2 similarity  |          82.7          |      81.0      |
> | STCN + Dot product   |          84.3          |      84.1      |
> | STCN + L2 similarity |          83.1          |      85.4      |
>
> **4\.** **Necessity of CBAM.** In a later experiment, we find that CBAM accelerates convergence but is not necessary for the final performance. With longer training, the performance is the same with/without CBAM.
>
> **5\.** **Giving up the value encoder.** It performs significantly worse. See also point 1.2.
>
> **6\.** **Without static image pretraining.** Many related works like STM [17], KMN [21], RMNet [25], MiVOS [20], GraphMem [15], etc. already employ static image pretraining. We believe using sufficient data encourages more expressive and powerful models that can infer patterns from data instead of relying on hand-crafted mechanisms. Notwithstanding, we can also provide results without pretraining in the main paper for reference.

---

### Decision · Program_Chairs · 2021-09-27

**Decision:**

Accept (Poster)

**Comment:**

This paper has four positive reviews (6,6,6,7). While the scores are close to the borderline, the reviewers are consistent, and appreciate the same points in the paper. It should be accepted to NeurIPS.